# Enzymatic Characterization of Unused Biomass Degradation Using the *Clostridium cellulovorans* Cellulosome

**DOI:** 10.3390/microorganisms10122514

**Published:** 2022-12-19

**Authors:** Mohamed Yahia Eljonaid, Hisao Tomita, Fumiyoshi Okazaki, Yutaka Tamaru

**Affiliations:** 1Department of Life Sciences, Graduate School of Bioresources, Mie University, 1577 Kurimamachiya, Tsu 514-8507, Japan; 2Department of Bioinfomatics, Mie University Advanced Science Research Center, Mie University, 1577 Kurimamachiya, Tsu 514-8507, Japan; 3Smart Cell Innovation Research Center, Mie University, 1577 Kurimamachiya, Tsu 514-8507, Japan

**Keywords:** cellulosic biomass, enzymatic degradation, cellulosome, *C. cellulovorans*

## Abstract

The cellulolytic system of *Clostridium cellulovorans* mainly consisting of a cellulosome that synergistically collaborates with non-complexed enzymes was investigated using cellulosic biomass. The cellulosomes were isolated from the culture supernatants with shredded paper, rice straw and sugarcane bagasse using crystalline cellulose. Enzyme solutions, including the cellulosome fractions, were analyzed by SDS-PAGE and Western blot using an anti-CbpA antibody. As a result, *C. cellulovorans* was able to completely degrade shredded paper for 9 days and to be continuously cultivated by the addition of new culture medium containing shredded paper, indicating, through TLC analysis, that its degradative products were glucose and cellobiose. Regarding the rice straw and sugarcane bagasse, while the degradative activity of rice straw was most active using the cellulosome in the culture supernatant of rice straw medium, that of sugarcane bagasse was most active using the cellulosome from the supernatant of cellobiose medium. Based on these results, no alcohols were found when *C. acetobutylicum* was cultivated in the absence of *C. cellulovorans* as it cannot degrade the cellulose. While 1.5 mM of ethanol was produced with *C. cellulovorans* cultivation, both n-butanol (1.67 mM) and ethanol (1.89 mM) were detected with the cocultivation of *C. cellulovorans* and *C. acetobutylicum*. Regarding the enzymatic activity evaluation against rice straw and sugarcane bagasse, the rice straw cellulosome fraction was the most active when compared against rice straw. Furthermore, since we attempted to choose reaction conditions more efficiently for the degradation of sugarcane bagasse, a wet jet milling device together with L-cysteine as a reducing agent was used. As a result, we found that the degradation activity was almost twice as high with 10 mM L-cysteine compared with without it. These results will provide new insights for biomass utilization.

## 1. Introduction

Lignocellulose is the most abundant raw material on the Earth. Its low price makes it an ideal feedstock for second generation biorefining aimed at replacing fossil-derived production of fuels and chemicals [1]. In order to utilize cellulosic materials, such as agricultural wastes, the consolidated bioprocessing (CBP) was performed with *Clostridium cellulovorans* and *C. beijerinckii* or methanogens [2,3]. Notably, the Carbohydrate-Active enZymes (CAZys) encoded by the *C. cellulovorans* genome were 37% higher than those found in *C. thermocellum*, one of the most efficient cellulolytic microorganisms reported until now [4]. A number of these enzymes—namely, those with a dockerin domain—physically associated to create enormous protein complexes, termed cellulosomes, that are attached to the cell surface of *C. cellulovorans*. 

The availability of the growth substrate(s) has been demonstrated in several studies to modulate the expression of *C. cellulovorans* cellulosomal and non-cellulosomal CAZys [5,6,7,8]. On the other hand, characterization of the enzymatic synergies in *C. cellulovorans* has been studied [9,10,11,12,13,14,15,16]. In particular, there exists a cellulosomal gene cluster in *C. cellulovorans,* which includes five glycosyl hydrolase family 9 (GH9) genes and a GH5 gene downstream of *cbpA* (the largest scaffold protein)-*exgS* (GH48) operon [3]. Among cellulosomal GH9 cellulases, such as EngH, EngK, EngL, EngM and EngY, EngK revealed a processive endo-type enzyme, whose products were only glucose and cellobiose from cello-oligosaccharides (G2-G6) as a substrate [17]. 

Therefore, synergistic effects for either EngH (GH9) or EngK (GH9) and ExgS (GH48) were investigated [16]. Whereas EngH and ExgS showed low synergy in the presence of 5 mg/mL cellobiose, EngK and ExgS (the ratio was EngK:ExgS = 75:25) had higher activity in even the same condition. More interestingly, only ExgS or EngH showed little activity in the present of 5 mg/mL cellobiose, while only EngK had enzymatic activity. Thus, enzymatic properties appear crucial for cellulosic biomass degradation.

One approach was performed as the conversion ability of alive plant cells to protoplasts using the culture supernatants from *C. cellulovorans,* which contained enzymatic activities for CMCase, xylanase, mannanase and pectate lyase [18]. These supernatants readily released protoplasts from cultured tobacco cells and *Arabidopsis thaliana*, with the result that the culture supernatant from pectin-grown cells was more active than supernatants from glucose-, cellobiose-, xylan- and locust-bean-gum-grown cells. Interestingly, the crude culture supernatant lost its protoplast formation activity after the removal of cellulosomes. 

Another approach was directly converted from degradative sugars to fermentable products, such as alcohols, organic acids and gases in combination with other fermentable anaerobes. In the cocultivation of *C. cellulovorans* and *C. beijerinckii*, IBE fermentation was performed using mandarin orange wastes [2]. Moreover, methane was produced from sugar beet pulp [3] and mandarin orange peel [19] under cocultivation with *C. cellulovorans* and methanogens.

In this study, we observed the long-term and continuous cultivation of *C. cellulovorans* by shredded paper as unused biomass. To optimize the reaction conditions with cellulosomes and non-cellulosomal enzymes for the degradation of unused various biomass, it was reported that three variables known to influence cellulosomal activity and stability were examined: oxygen sensitivity, stabilization by calcium and cellobiose inhibition [20]. In this paper, optimal cellulosomal activity was achieved when digestions contained L-cysteine as a reducing agent, CaCl_2_ for stabilization and b-D-glucosidase to mitigate cellobiose inhibition. 

Therefore, we investigated the enzymatic degradation of rice straw and sugarcane bagasse with the isolated cellulosomes from *C. cellulovorans,* which were cultivated with cellobiose, rice straw and sugarcane bagasse, respectively. In addition, in order to perform more efficient degradation for sugarcane bagasse, both pretreatment by a wet jet milling device and enzymatic reaction conditions with L-cysteine were investigated.

## 2. Materials and Methods

### 2.1. Culture Conditions

*Clostridium cellulovorans* 743B (ATCC35296) was grown anaerobically as described previously [18], with the exception of a carbon source in the media. As carbon sources, 1% (*w*/*v*) cellobiose (Sigma-Aldrich, Tokyo, Japan), 1% (*w*/*v*) shredded paper (Mie University), 1 % (*w*/*v*) rice straw (Mie, Japan), or 1% (*w*/*v*) sugarcane (Okinawa, Japan) were used. *C. acetobutylicum* ATCC 824 was cultivated in the same medium as *C. cellulovorans*. After the cultivation of *C. cellulovorans* for 7 days, *C. acetobutylicum* was inoculated and cultivated for 7 days.

### 2.2. Preparation and Purification of the Cellulosome Fraction

After the cultivation of *C. cellulovorans* with carbon sources, the supernatants were centrifuged at 5000× *g*, at 4 °C for 20 min and were filtered with 0.45 mm DISMIC AS (Advantec Toyo Kaisha, Ltd., Tokyo, Japan), and the precipitate was washed twice with Milli-Q water using a glass filter. Both fractions were precipitated with 90% ammonium sulfate and then were centrifuged at 15,000× *g*, at 4 °C for 30 min. The precipitates were dialyzed with 50 mM phosphate buffer (pH7.0). Further purification was performed by crystalline cellulose (Avicel PH-101, Sigma-Aldrich, Tokyo, Japan). The adsorbed fraction to crystalline cellulose was eluted with distilled water.

### 2.3. SDS-PAGE and Western Blot Analysis

Sodium dodecyl sulfate-polyacrylamide gel electrophoresis (SDS-PAGE) was performed on a 7.5% polyacrylamide gel by the method of Laemmli [21]. After electrophoresis, the gel was stained with Coomassie brilliant blue R. Western blot analysis was performed using an anti-CbpA antiserum (diluted 1:2000). The procedure was conducted as described previously [22].

### 2.4. Enzymatic Activity, Protein Assays and TLC Analysis

The reaction mixture consisted of 1 mL of enzyme solution, 5 mL of 1% rice straw or 1% sugarcane and 4 mL of 100 mM phosphate buffer (pH 7.0). The mixture was incubated at 37 °C, and then the reducing sugar as measured by cellulase activity was measured using a microplate-based dinitrosalicylic acid (DNS) method. One unit of enzyme activity was defined as the amount of enzyme that liberates 1 μmol of D-glucose (FUJIFILM Wako Pure Chemical Corp., Osaka Japan) per min under the above conditions. 

The protein concentrations were measured by the method of Bradford with a protein assay kit from Bio-Rad, using bovine serum albumin as a standard. Thin-layer chromatography (TLC) analysis was performed as described previously [22]. Purified cellulosomes were incubated with for 16 h at 37 °C. The reaction products were separated on TLC plates (Merck KGaA, Darmstadt, Germany) with a solvent system containing 1-butanol–ethanol–water (5:5:2.5). For the detection of the products, the plate was sprayed with staining reagent (5% H_2_SO_4_ in methanol) and baked for 10 min at 100 °C.

### 2.5. Alcohol Concentration

Alcohol concentrations, such as ethanol and n-butanol were measured by a gas chromatograph GC-2010plus (Shimadzu, Kyoto, Japan) with a capillary column Rt-Q-BOND (30 m, inner diameter. 0.32 mm; RESTEK Corp., Bellefonte, PA, USA). The oven temperature was 250 °C, and the column temperature was 150 °C. Nitrogen was the carrier gas, which was set at a flow rate of 1.21 mL/min.

### 2.6. Pretreatment of Sugarcane Bagasse and Its Enzymatic Reaction

We pretreated 2% (*w*/*v*) sugarcane bagasse using a wet jet milling device under 200 MPa (Star Bust^®^, SUGINO Machine Ltd., Toyama, Japan) from 1 to 30 passes (Figure 1). The reaction mixture consisted of 6 mL of enzyme solution cultivated with untreated or treated sugarcane bagasse and cellobiose, 10 mL of 5% sugarcane bagasse without pretreatment, 4 mL of 250 mM acetate buffer (pH 6.0), 0.16 mL of 10 mg/mL tetracycline solution (Sigma-Aldrich No. T7660, Tokyo, Japan) in 70% ethanol and 0.12 mL of 10 mg/mL cycloheximide solution (Sigma-Aldrich No. C1988, Tokyo, Japan) in distilled water. The mixture was incubated at 37 °C, and then the reducing sugar as D-glucose was measured using the DNS method. The data represent at least three independent experiments.

## 3. Results

### 3.1. Continuous Cultivation of C. cellulovorans with Shredded Paper

*C. cellulovorans* was cultivated with shredded paper as a carbon source. SDS-PAGE revealed that several bands between the non-adsorption and adsorption bands were different (Figure 2A, lanes 2 and 3). As shown in Figure 2B, Western blot analysis using anti-CbpA antibody showed the bands of CbpA in lanes 1 and 3, indicating that the cellulosomal proteins were recovered by crystalline cellulose. 

TLC analysis showed the main products in the culture supernatant were detected as glucose and cellobiose after the shredded paper was completely degraded for 9 days (Figure 2C). According to continuous cultivation with shredded paper, the shredded paper was complete degraded by *C. cellulovorans* without any pretreatment. Furthermore, the concentration of the reducing sugar was gradually increased and accumulated to reach 1.46 mg/mL at 30 days, and shredded paper was added at a final concentration of 1% shredded paper in replacement of half the volume of fresh culture medium (Figure 3).

### 3.2. Cocultivation of C. cellulovorans and C. acetobutylicum

*C. cellulovorans* was cultivated with shredded paper for 7 days, and then *C. acetobutylicum* was inoculated into the same medium for 7 days. As a result, both n-butanol (1.67 mM) and ethanol (1.89 mM) were detected with co-cultivation of *C. cellulovorans* and *C. acetobutylicum*, while ethanol was detected with only the cultivation of *C. cellulovorans* (Table 1). No alcohols were detected in the supernatant of *C. acetobutylicum* with shredded paper. These results suggested that *C. acetobutylicum* could produce n-butanol and ethanol using glucose and cellobiose after the degradation of shredded paper by *C. cellulovorans* (Figure 2C).

### 3.3. Cultivation of Cellulosic Biomass with C. cellulovorans

*C. cellulovorans* was cultivated with cellulosic biomass, such as sugarcane bagasse and rice straw. After 72 h cultivation, the cellulosome fractions purified by crystalline cellulose were prepared from the supernatants. Western blotting was performed with anti-CbpA antibodies, with the results that CbpA bands (170 kDa) were detected from all fractions (Figure 4). Thus, *C. cellulovorans* produced the cellulosomes introduced by sugarcane bagasse and rice straw as carbon sources, respectively.

By using the cellulosome fractions from cellobiose, sugarcane bagasse and rice straw, enzymatic activities were measured against rice straw and sugarcane bagasse (Figure 5). As a result, the cellulosome fraction from rice straw was the most active compared with the others (Figure 5A). On the other hand, the cellulosome fraction from cellobiose had the highest activity over the others (Figure 5B). These results indicated that sugarcane bagasse could be degraded by the cellobiose cellulosome, which could be prepared from shredded paper. In contrast, the cellulosome from sugarcane bagasse seemed to contain different enzyme subunits compared to the rice straw cellulosome. Therefore, the cellobiose cellulosome might be also able to be used for rice straw degradation.

### 3.4. Futher Analysis of Sugarcane Bagasse Degradation

In order to promote degradation of sugarcane bagasse, pretreatment was performed using a wet jet milling device. 2% (*w*/*v*) sugarcane bagasse was treated from untreated to 30 passes. The medium was prepared at a final concentration of 1% sugarcane bagasse with untreated and 10-pass treated, and then the cultivation of *C. cellulovorans* was performed. 

Through SDS-PAGE, the band patterns between untreated and treated sugarcane bagasse appeared to be similar to each other, while those of the cellobiose supernatant were different from the others (Figure 6A). Western blot analysis revealed that CbpA (170 kDa) was detected among all supernatants (Figure 6B), suggesting the cellulosomes were clearly produced in the supernatants. The proteins in the supernatants were precipitated by ammonium sulfate and then were concentrated and dialyzed by 100 mM phosphate buffer (pH6.0) as enzyme solutions. 

Next, the enzyme reaction was performed with 5% (*w*/*v*) sugarcane bagasse treated with 30 passes as a substrate at 37 °C, together with 200 rpm rotation admixture. As the result of reducing sugar measurement, enzyme solution from cellobiose supernatant (E3) was most active and the final activity had 1.4-times higher than that of pretreated sugarcane bagasse supernatant (E1) (Figure 7). In contract, the enzyme solution of untreated sugarcane bagasse supernatant (E2) was a little active than that of the treated one. As shown in Table 2, specific activity (38.2 mU/mg) in E2 was higher than those in E1 (27.9 mU/mg) and E3 (28.0 mU/mg), respectively. 

These results indicated E3 (cellobiose supernatant) was most active against 30-pass-pretreated sugarcane bagasse as a substrate, even though the specific activity of E3 (cellobiose supernatant) was lower than that of E2 (untreated sugarcane bagasse supernatant). More interestingly, the degradation activity of E2 was higher than that of E1 (Figure 7), whereas the specific activity of E2 was 1.37-times higher than that of E1 (Table 2). These results suggest that *C. cellulovorans* might produce key enzymes for the degradation of pretreated sugarcane bagasse.

### 3.5. Effect of Enzymatic Reaction of Sugarcane Bagasse with L-Cysteine

In order to promote the degradation of sugarcane bagasse by *C. cellulovorans* and to confirm whether the pretreatment with a wet jet milling device had synergistic effects for degradation, L-cysteine was used for the enzymatic reaction with E3. The reaction mixture was contained at a final concentration of 10 mM L-cysteine. As a result, degradation activity with 10 mM L-cysteine was approximately twice as high as that without L-cysteine (Figure 8). 

Furthermore, degradative activities by varying the number of pretreatments were compared with the addition of 10 mM L-cysteine (Figure 9). These results suggested that 30-pass-treated sugarcane bagasse had the highest activity, and its degradative activity was 1.4-times higher than the one-pass-treated sample. Interestingly, degradative activity with untreated sugarcane bagasse was finally reached for both the three-pass and five-pass samples. Finally, the total of degradative activities on sugarcane bagasse was at least three-times higher than without anything.

## 4. Discussion

In the last two decades, there has been a significant emphasis in sustainable, biological methods as a response to the escalating energy and climate crises. In contrast to producing biofuels from edible polysaccharides, such as starch, the current approaches have concentrated on next-generation bioenergy to generate higher-energy fuels from inedible lignocellulosic biomass, which exist in a range of copious plant materials [23]. The enzymatic hydrolysis of biomass produces sugars, which may be transformed into a wide variety of biofuels and consumables. 

On the other hand, enzymes represent a significant operating cost in the cellulosic bioenergy process [24]. As hydrolyzing cellulose is more difficult compared with starch, the development of more potent enzyme combinations is necessary. In addition, to boost sugar yield and reduce enzyme differences amongst biomass resources, broad-specificity glycoside hydrolases (GHs) can supplant dozens of specialized enzymes with fewer, more flexible catalysts, increasing the sugar yield and reducing the enzyme variability between feedstocks [25]. Thus far, around 173 GH families have been identified (accessed on 1 December 2022, http://www.cazy.org/Glycoside-Hydrolases.html).

A total of 57 cellulosomal genes were found in the *C. cellulovorans* genome and coded for not only carbohydrate-active enzymes but also lipase, peptidase and proteinase inhibitors, in addition to two novel genes encoding scaffolding proteins CbpB and CbpC [26]. Among 57 cellulosomal genes, 53 dockerin-containing proteins and four cohesin-containing scaffolding proteins were comprised [27]. 

Three scaffolding proteins, the main CbpA and newly found CbpB and CbpC that are tandemly localized in the *C. cellulovorans* genome, consisting of a carbohydrate-binding module (CBM) of family 3, a surface–layer homology domain and a cohesin domain, have been identified. While 18 cellulases consist of endoglucanases belonging to families 5 (GH5) and 9 (GH5) and a family 48 exoglucanase (GH48), 10 hemicellulases comprise six mannanases belonging to families 5 (GH5) and 26 (GH26), three xylanases belonging to families 8 (GH8) and 10 (GH10) and a family 98 endo-b-galactosidase (GH98). In addition, two cellulosomal pectate lyases belonging to families 1 (PL1) and 9 (PL9) have been found in the *C. cellulovorans* genome. 

Whereas whole-cell proteomes of *C. cellulovorans* grown on either glucose- or cellulose-supplemented medium were compared for a total of 1016 identified proteins in both conditions [28], about 15% of the proteins in *C. cellulovorans*, which equals 621 proteins, were detected [29]. Several glycolytic enzymes, fermentation pathways (such as hydrogenase, pyruvate formate lyase and phosphate transacetylase) and nitrogen assimilation (such as glutamate dehydrogenase) were all modulated by growth substrates. Additionally, it appears that *C. cellulovorans* fed on cellulose expends more energy, which appears to be connected to the overexpression and secretion of hemicelluloses. High energy consumption drives the up-regulation of ATP synthesis pathways (such as ATP synthase and acetate production) [29].

Utilizing vaster and more varied feedstock, including the plentiful renewable resources of lignocellulosic biomass found in nature, is made possible by co-culturing *Clostridium* spp. and other microorganisms. Examples of the lignocellulosic biomass include cedar [30], aspen [31], agave [32], cassava [33], miscanthus biomass [34], switchgrass [35] and salix [36] as well as various agricultural residues. It seems possible to use fruit residue and to harvest remaining straw and industrial remnants, such as yeast waste and biodiesel waste (crude glycerol) [37]. The production of extracellular cellulosomes by *C. cellulovorans* allows for the effective digestion of native soft biomass materials, such as corn cobs [10,38,39], sugar beet pulp [3], orange wastes [2,19] and rice thatch [26]. 

In this study, cellulose was sourced from shredded paper, and the lignocellulose was from rice straw and sugarcane bagasse. Shredded paper’s chopped fibers make it challenging to easily reconstitute into high-quality paper goods [40]. Additionally, short strips or pieces of paper cannot be recycled at the majority of facilities. Large-scale recycling facilities use large screens to dry pulped paper on, and finely shredded paper is not well retained and can fall through the screens. Furthermore, since cellobiose is much more expensive than glucose, *C. cellulovorans* can prepare it from shredded paper (Figure 2C). 

Since enzyme solution from the supernatant of cellobiose cellulosome (E3) were most active against sugarcane bagasse (Figure 5C and Figure 6), it is easy to obtain active cellulosomes by the combination of shredded paper with other cellulosic wastes. In addition, we found that L-cysteine as an additive increased enzymatic activity for degradation of sugarcane bagasse (Figure 7). The synergistic effects of fungal free enzymes and cellulosomes from *C. thermocellum* on Avicel were examined by activity assays [20]. 

For the combination of enzyme systems, reaction conditions were chosen wherein the activity of both systems maintained at least 90% of their optimal activity—namely, at 50 °C in 30 mM sodium acetate pH 5.5 buffer containing 10 mM CaCl_2_, 100 mM NaCl, 2 mM EDTA and 10 mM L-cysteine. As a result, the combination of cellulosomes and free enzymes exhibited the highest activity on Avicel tested, reaching 100% conversion in 24 h in this study. In case of other hydrolytic enzyme, such as papain, the effects of cysteine concentration in the range of 0–40 mM on the specific activity of free and both papain immobilized forms were compared [41]. 

As a result, it appeared that the maximum activity of the simple immobilized enzyme was found in the presence of 10 mM L-cysteine. To begin with, since the *C. cellulovorans* medium contains 0.1% L-cysteine, approximately 5.6 mM L-cysteine is absent in the medium. These results suggested that in the absence of 10 mM L-cysteine its specific activity is even higher for the cellulose than free enzymes, such as non-cellulose components. On the other hand, physical crushing by a wet jet milling device was sufficiently efficient to hydrolyze 30-pass-treated sugarcane bagasse (Figure 8). Thus, these results provide important information for future biomass utilization. 

## 5. Conclusions

In this study, we investigated the cultivation of *C. cellulovorans* to obtain degradative enzymes for unused biomass (such as shredded paper, sugarcane bagasse and rice straw) with reaction conditions using the addition of 10 mM L-cysteine and pretreatment of biomass with a wet jet milling device. Enzyme solutions containing the cellulose prepared from the supernatant of cellobiose medium (E3) from *C. cellulovorans* were more active compared with the other enzyme solutions. More interestingly, since glucose and cellobiose were the main products obtained from shredded paper, a combination of cellulosic biomass for celluloses and non-cellulose enzymes would be more useful for their enzyme preparation containing active cellulose from *C. cellulovorans*.

## Figures and Tables

**Figure 1 microorganisms-10-02514-f001:**
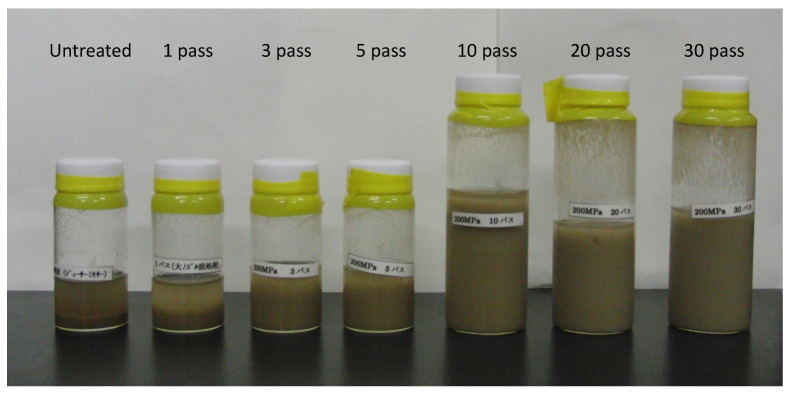
Pretreatment of 2% sugarcane bagasse with the wet jet milling device. 2% (*w/v*) sugarcane bagasse was pretreated using the wet jet milling device under 200 MPa (Star Bust®, SUGINO Machine Ltd., Toyama, Japan) from 1 to 30 passes and without it.

**Figure 2 microorganisms-10-02514-f002:**
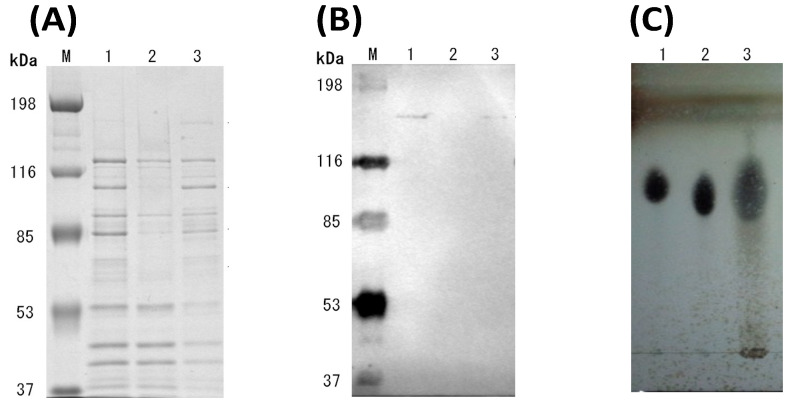
Analyses of the culture supernatant of *C. cellulovorans* with shredded paper. (**A**) SDS-PAGE and (**B**) Western blot analysis using an anti-CbpA antibody (1:2000 dilution). Lane M, molecular mass marker; lane 1, culture supernatant; lane 2, non-adsorption fraction against shredded paper; and lane 3, adsorption fraction against shredded paper. (**C**) TLC analysis. Lane 1, glucose (G1); lane 2, cellobiose (G2); and lane 3, culture supernatant.

**Figure 3 microorganisms-10-02514-f003:**
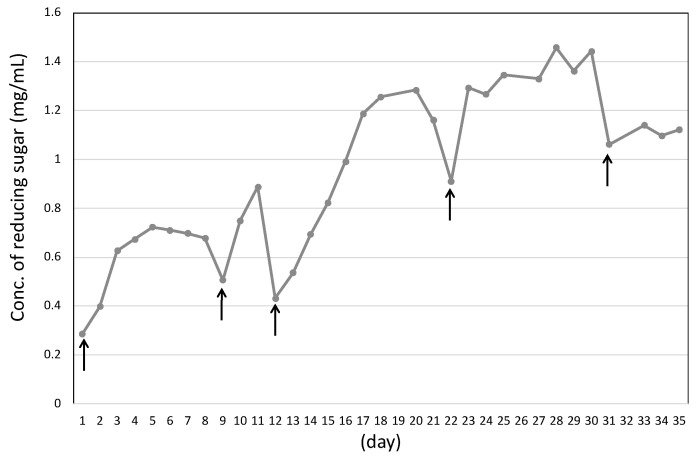
Measurement of reducing sugar in the culture supernatant of *C. cellulovorans* with shredded paper. Arrows indicate where the shredded paper was added after it was completely degraded.

**Figure 4 microorganisms-10-02514-f004:**
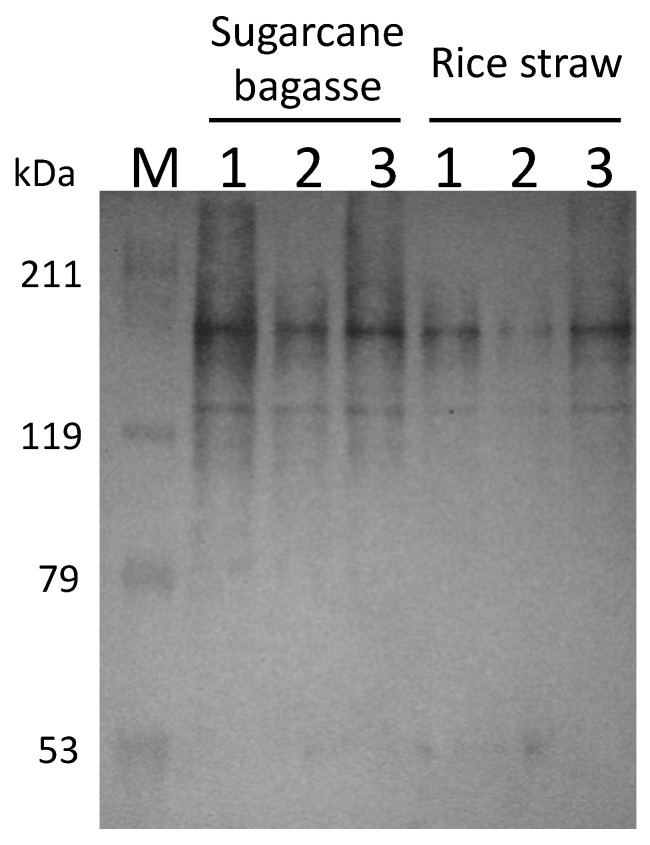
Western blot analysis of the supernatant and adsorption or non-adsorption fractions to crystalline cellulose from sugar bagasse and rice straw media. Lane M: Protein mass marker; lane 1: supernatant fraction; lane 2: non-adsorption fraction with Avicel; lane 3: adsorption fraction with Avicel. The anti-CbpA antibody was used as 1:2000 dilution.

**Figure 5 microorganisms-10-02514-f005:**
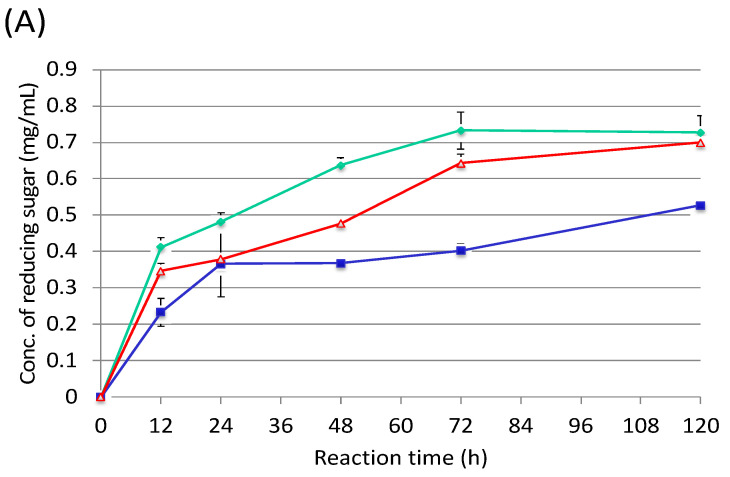
Enzymatic activities of rice straw (**A**) and sugarcane bagasse (**B**) using each purified cellulosome. Red line, cellobiose cellulosome; green line, rice straw cellulosome; blue line, sugarcane baggase cellulosome.

**Figure 6 microorganisms-10-02514-f006:**
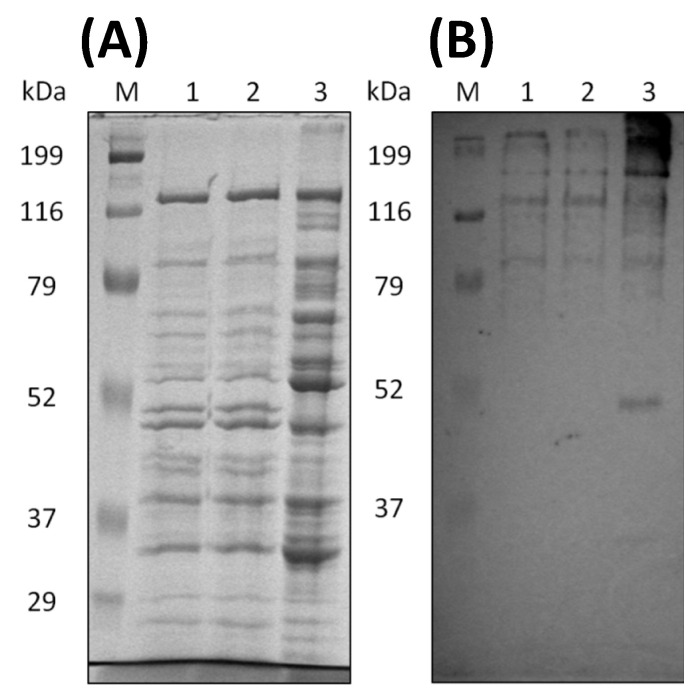
Analyses of the culture supernatants of *C. cellulovorans* with cellobiose and untreated or treated sugarcane bagasse. (**A**) SDS-PAGE and (**B**) Western blot analysis using an anti-CbpA antibody (1:2000 dilution). Lane M, molecular mass marker; lane 1, supernatant cultivated with 1% pretreated sugarcane bagasse, lane 2: supernatant cultivated with 1% untreated sugarcane bagasse; lane 3, supernatant cultivated with 0.5% cellobiose.

**Figure 7 microorganisms-10-02514-f007:**
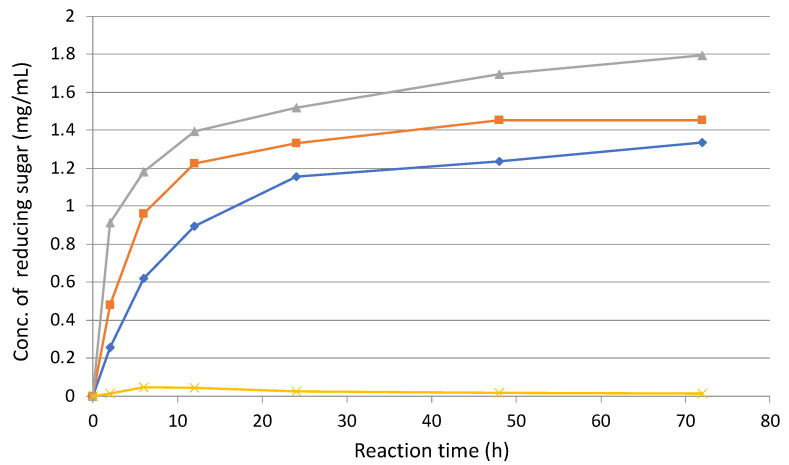
Enzymatic activities of 30-pass-treated sugarcane bagasse as a substrate using each enzyme solution. Enzyme solutions: E1, enzyme solution from pretreated sugarcane bagasse medium (rhombus); E2, enzyme solution from untreated sugarcane bagasse medium (square); and E3, enzyme solution from cellobiose medium (triangle). Negative control (cross).

**Figure 8 microorganisms-10-02514-f008:**
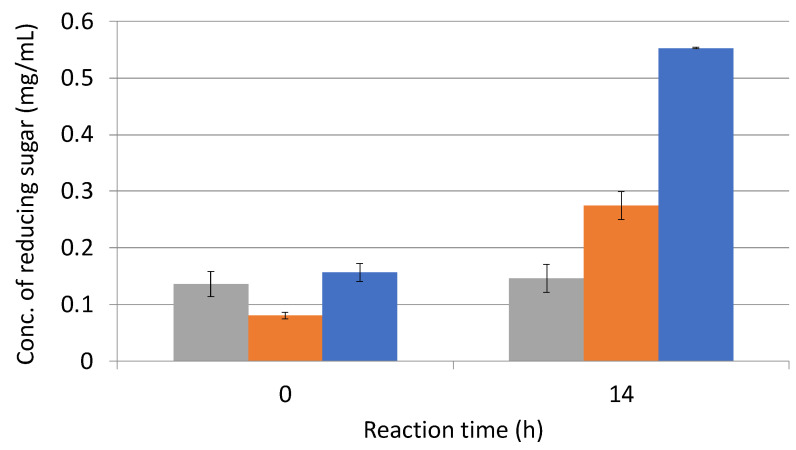
Enzymatic activities of 30-pass-treated sugarcane bagasse with L-cysteine. Grey bar, negative control with only phosphate buffer; orange bar, without L-cysteine; and blue bar, with 10 mM L-cysteine. A negative control was used with distilled water instead of enzyme solution.

**Figure 9 microorganisms-10-02514-f009:**
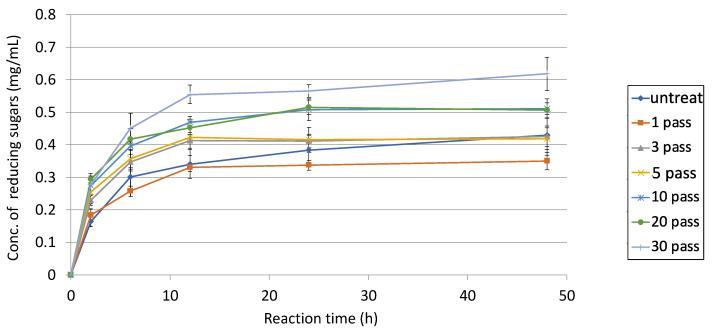
Degradative activities of untreated and 1- to 30-pass-treated sugarcane bagasse using E3 with 10 mM L-cysteine. 2% (*w*/*v*) sugarcane bagasse was pretreated by the wet jet milling device. After the pretreatment from 1 to 30 passes with the device or without it, all samples were used as a substrate for enzymatic reaction. D-Glucose was used as a standard curve for reducing sugars. Each line indicates each treatment.

**Table 1 microorganisms-10-02514-t001:** Concentrations of ethanol and n-butanol from the different cultivations with shredded paper.

Bacterial Cultivation	Concentration (mM)	
	Ethanol	n-Butanol
*C. cellulovorans* *^1^	1.50	0
*C. acetobutylicum* *^1^	0	0
Cocultivation *^2^	1.89	1.67
Negative control	0	0

*C. cellulovorans* was cultivated with the medium containing shredded paper for 7 days. After that, *C. acetobutylicum* was inoculated into the same medium and was cultivated for 7 days. The concentration of alcohols was measured using gas chromatography. *^1^, 7 days of cultivation; *^2^, 14 days of cultivation.

**Table 2 microorganisms-10-02514-t002:** Degradation activity of 30-pass-treated sugarcane bagasse for 12 h reaction.

Enzyme Solution	Degradation Activity with Pretreated Sugarcane Bagasse (mU/mL)	Final Protein Concentration (mg/mL)	Specific Activity (mU/mg)
E1	6.90	0.247	27.9
E2	9.44	0.247	38.2
E3	10.8	0.385	28.0

E1, enzyme solution from pretreated sugarcane bagasse medium; E2, enzyme solution from untreated sugarcane bagasse medium; E3, enzyme solution from cellobiose medium.

## Data Availability

Not applicable.

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
