# Peer review of "Enzymatic Characterization of Unused Biomass Degradation Using the *Clostridium cellulovorans* Cellulosome"

_microorganisms, 2022, doi:10.3390/microorganisms10122514_

Round 1
Reviewer 1 Report
The manuscript entitled “Enzymatic Characterization of Unused Biomass Degradation Using the Clostridium cellulovorans Cellulosome” by M. Y. Eljonaid and coauthors, describes interesting experiments of cultivations of the cellulolytic and mesophilic bacterium on various cellulosic biomasses derived from agricultural byproducts and paper wastes for future valorization. Analyses of the cellulosomes fractions were also performed, and showed various degradative activities, the “cellobiose” derived cellulosomes being the most efficient.
Altogether, the reported data are of interest. However, the authors should comment further the fact that cellobiose-derived cellulosomes were unexpectedly the most efficient on nearly all types of cellulosic biomass.
My main concern about this manuscript relates to the English which must be significantly improved. For instance, rice straw is often spelt “rice strow”. I strongly advice the authors to make their manuscript reviewed by a native English-speaking scientist.
Besides some sentences are unclear, for instance lanes 21-24 (end of abstract) or lanes 71-74. Others can be omitted like the sentence at lanes 53-53.
Also, the authors often confuse a gene and its product. For instance in lane261, “57 cellulosomal genes” should be replaced by “57 genes encoding cellulosomal components” etc…
Minor issues
Figure 7, legend please explain “negative control”
Lanes : 198-199, acetate is not an appropriate buffer for pH 6.0 (pKa = 4.5!)
Throughout the materials and methods section, “g” should be in italics.
The quality of figures 6 and 8 should be improved (thicker line )
Author Response
Dear Reviewer #1:
Thanks a lot for your comments and suggestions. We deeply appreciate you for taking care of them. Please see the below.
Best regards,
Prof. Dr. Yutaka TAMARU
My main concern about this manuscript relates to the English which must be significantly improved. For instance, rice straw is often spelt “rice strow”. I strongly advise the authors to make their manuscript reviewed by a native English-speaking scientist.
--> We ordered English check for English proofreading that the journal recommended.
Besides some sentences are unclear, for instance lanes 21-24 (end of abstract) or lanes 71-74. Others can be omitted like the sentence at lanes 53-53.
--> The sentence at lanes 53-53 was deleted, and the other sentences pointed out were modified, respectively.
For lanes 21-24, the sentence was modified as “in order to perform more efficient degradation for sugarcane bagasse, a wet jet milling device was used together with L-cysteine as an additive. As a result, the degradative activity of pretreated sugarcane bagasse was increased by active cellulosome which was prepared from the supernatant of cellobiose medium. Thus, these results will provide new insights for biomass utilization.”
For lanes 71-74, the sentence was modified as “in order to perform more efficient degradation for sugarcane bagasse, pretreatment by a jet milling device and enzymatic reaction condition with L-cysteine were investigated, respectively.”
Also, the authors often confuse a gene and its product. For instance in lane 261, “57 cellulosomal genes” should be replaced by
--> We corrected it to “57 genes encoding cellulosomal components” etc…
Minor issues
Figure 7, legend please explain “negative control”
--> Negative control is no enzymatic reaction with only phosphate buffer.
Lanes : 198-199, acetate is not an appropriate buffer for pH 6.0 (pKa = 4.5!)
--> We corrected it to the phosphate buffer (pH6.0).
Throughout the materials and methods section, “g” should be in italics.
--> We corrected to italics “g”.
The quality of figures 6 and 8 should be improved (thicker line)
--> We improved Figures 6 and 8, respectively.
Reviewer 2 Report
The study describes the characterization of the cellulosomes isolated from the culture supernatants with shredded paper, rice straw, and sugarcane bagasse. The authors also investigated the enzyme reaction conditions of sugarcane bagasse those enzyme solutions. In addition, the authors used L-cysteine as an additive to increase degradative activity of sugarcane bagasse. However, the objective of using L-cysteine was not clear as well as the cocultivation strategy.
1. A deep review of the manuscript is needed. There are lot of mistakes such as “degrative” instead of “degradative” (line 22) and “was” instead of “were” (line 312).
2. In the abstract, the sentence “According to rice straw and sugarcane bagasse,…” (line 18-21) does not make sentence. “Regarding the rice straw and sugarcane bagasse,…” would be better.
3. Describe CAZys when it appears at the first time (line 34).
4. In the introduction, the second paragraph does not describe relevant data for this study. It seems to be written to provide citations of authors’ previous publications. It does not make clear the reason of cocultivation strategy in this study. The introduction should explore subjects that are important to understand and justify this study such as the different types of biomass used, the use of L-cysteine, the applications of cellulosome, among others.
5. “sugar” instead of “suger” (line 151)
6. Is the sentence “Since no alcohols were detected…” (line 158-161) correct? I understand that C. acetobutylicum cannot produce alcohols from glucose and cellobiose since they were not detected when this microorganism was cultivated in the absence of C. cellulovorans. Please, correct this sentence and improve the discussion of these results. Why did butanol only appear in the cocultivation? Why did the authors study the cocultivation? Why did the authors choose the C. acetobutylicum?
7. Which alcohols were investigated (line 113)?
8. Regarding the Table 1, the caption should be more detailed. These concentrations were obtained after 7 or 14 days? The cocultivation lasted 14 days, right?
9. Replace the sentence “In contrast, 179 the cellulosome from sugarcane bagasse seemed to contain different enzyme subunits from the rice straw cellulosome.” by “In contrast, 179 the cellulosome from sugarcane bagasse seemed to contain different enzyme subunits compared to the rice straw cellulosome.”
10. The discussion about the use of L-cysteine should be improved. What happens in the presence of the L-cysteine?
11. Please correct “poly-saccharides” (line 250).
12. The conclusions must be improved.
Author Response
Dear Reviewer #2:
Thanks a lot for your comments and suggestions. We deeply appreciate you for taking care of them. Please see the below.
Best regards,
Prof. Dr. Yutaka TAMARU
1. A deep review of the manuscript is needed. There are lot of mistakes such as “degrative” instead of “degradative” (line 22) and “was” instead of “were” (line 312).
--> We corrected to “degradative” and “were”, respectively.
2. In the abstract, the sentence “According to rice straw and sugarcane bagasse,…” (line 18-21) does not make sentence. “Regarding the rice straw and sugarcane bagasse,…” would be better.
--> We corrected to “Regarding the rice straw and sugarcane bagasse,…”
3. Describe CAZys when it appears at the first time (line 34).
--> We added “CAZys (Carbohydrate-Active enZymes)”
4. In the introduction, the second paragraph does not describe relevant data for this study. It seems to be written to provide citations of authors’ previous publications. It does not make clear the reason of cocultivation strategy in this study. The introduction should explore subjects that are important to understand and justify this study such as the different types of biomass used, the use of L-cysteine, the applications of cellulosome, among others.
--> modified to the second paragraph “In order to utilize cellulosic materials such as agricultural wastes, the consolidated bioprocessing (CBP) was carried out with Clostridium cellulovorans and C. beijerinckii or methanogens [2,3].” References 2 and 3 were replaced, respectively.
5. “sugar” instead of “suger” (line 151)
--> We corrected to “sugar”
6. Is the sentence “Since no alcohols were detected…” (line 158-161) correct? I understand that C. acetobutylicum cannot produce alcohols from glucose and cellobiose since they were not detected when this microorganism was cultivated in the absence of C. cellulovorans. Please, correct this sentence and improve the discussion of these results. Why did butanol only appear in the cocultivation? Why did the authors study the cocultivation? Why did the authors choose the C. acetobutylicum?
--> Yes, when C. cellulovorans was absent in the coculture, C. acetobutylicum never produced any alcohols such as ethanol and n-butanol. Although only C. cellulovorans produced ethanol in the culture supernatant from shredded paper, this microorganism never produces n-butanol. Since shredded paper was degraded by C. cellulovorans to produce glucose and cellobiose, the coculture of C. acetobutylicum with C. cellulovorans was performed to clarify whether C. acetobutylicum could produce n-butanol. Therefore, the sentence was corrected as “No alcohols were detected in the supernatant of C. acetobutylicum, when this microorganism was cultivated in the absence of C. cellulovorans.”
7. Which alcohols were investigated (line 113)?
--> Alcohols indicated ethanol and butanol. Therefore, the sentence was corrected as “Alcohol concentrations such as ethanol and n-butanol were measured by …”
8. Regarding the Table 1, the caption should be more detailed. These concentrations were obtained after 7 or 14 days? The cocultivation lasted 14 days, right?
--> First, C. cellulovorans was cultivated for 7 days and then C. acetobutylicum was inoculated into the same medium for another 7days. Therefore, the total of cultivation time was 14 days. The caption was added as “C. cellulovorans was cultivated with the medium containing shredded paper for 7 days. After that, C. acetobutylicum was inoculated into the same medium and was cultivated for 7 days. The concentration of alcohols was measured by gas chromatography.”
9. Replace the sentence “In contrast, 179 the cellulosome from sugarcane bagasse seemed to contain different enzyme subunits from the rice straw cellulosome.” by “In contrast, 179 the cellulosome from sugarcane bagasse seemed to contain different enzyme subunits compared to the rice straw cellulosome.”
--> We replaced to “In contrast, 179 the cellulosome from sugarcane bagasse seemed to contain different enzyme subunits compared to the rice straw cellulosome.”
10. The discussion about the use of L-cysteine should be improved. What happens in the presence of the L-cysteine?
--> We added a paper of protease to explain the mechanism and the reason.
11. Please correct “poly-saccharides” (line 250).
--> We corrected to “polysaccharides”
12. The conclusions must be improved.
--> We improved the conclusions.
Round 2
Reviewer 2 Report
1. Abstract should report the main results.
2. “degrative” is still written in the manuscript (now in lines 19, 60, 248, 250, 251, 252 and 260). “were” instead of “was” is not in line 316, it is in Conclusions (line 335). Please, correct the lines 316 and 335.
3. The introduction does not explain the reason of the use of L-cysteine. Why did authors investigate this?
4. If C. acetobutylicum never produced any alcohols such as ethanol and n-butanol why did the authors write “These results suggested C. acetobutylicum could produce n-163 butanol and ethanol using glucose and cellobiose from shredded paper” (lines 163-164)?
5. Regarding the Table 1, I suggest that the authors add the days of each cultivation using 1 or 2 for 7 or 14 days, respectively. The caption could be “Concentrations of ethanol and butanol from the different cultivations with shredded paper”
6. The discussion about the use of L-cysteine is still missing. I did not find the paper about protease and the explanation of the mechanism and the reasons.
7. The conclusions was not improved.

Author Response
Dear Reviewer #2:
Thank you very much for your kind suggestions.
We further corrected and modified our manuscript. Furthermore, we added the reference for the use of L-cysteine.
Best regards,
Dr. Yutaka Tamaru
- Abstract should report the main results.
--> We modified the abstract described in only the main results.
- “degrative” is still written in the manuscript (now in lines 19, 60, 248, 250, 251, 252 and 260). “were” instead of “was” is not in line 316, it is in Conclusions (line 335). Please, correct the lines 316 and 335.
--> We corrected all “degrative” to “degradative”.
- The introduction does not explain the reason of the use of L-cysteine. Why did authors investigate this?
--> We explained the use of L-cysteine in the Introduction. According to the paper of Energy Environ. Sci., 6, 1858 (2013), optimal cellulosomal activity was achieved when digestions contained L-cysteine as a reducing agent, CaCl2 for stabilization, and b-D- glucosidase to mitigate cellobiose inhibition. Therefore, we used L-cysteine in this research.
- If C. acetobutylicum never produced any alcohols such as ethanol and n-butanol why did the authors write “These results suggested C. acetobutylicum could produce n-163 butanol and ethanol using glucose and cellobiose from shredded paper” (lines 163-164)?
--> We would like to say that C. acetobutylicum never produced any alcohols such as ethanol and n-butanol with shredded paper as the sole carbon source, suggesting C. acetobutylicum was not able to degrade shredded paper to produce glucose and cellobiose. Therefore, we wrote “These results suggested C. acetobutylicum could produce n-butanol and ethanol using glucose and cellobiose from shredded paper”. So, ethanol and n-butanol were detected with co-cultivation of C. cellulovorans and C. acetobutylicum, while only ethanol was detected with the C. cellulovorans cultivation with shredded paper. However, we modified this sentence.
- Regarding the Table 1, I suggest that the authors add the days of each cultivation using 1 or 2 for 7 or 14 days, respectively. The caption could be “Concentrations of ethanol and butanol from the different cultivations with shredded paper”
--> We corrected the caption of Table 1 to “Concentrations of ethanol and butanol from the different cultivations with shredded paper”.
- The discussion about the use of L-cysteine is still missing. I did not find the paper about protease and the explanation of the mechanism and the reasons.
--> The reason why the use of L-cysteine was used was added in the discussion based on the research paper related to the reference [20]. In case of papain, the mechanism might be same as cellulases.
- The conclusions was not improved.
--> We modified the conclusions.
